# Fairness in Streaming Submodular Maximization: Algorithms and Hardness

**Marwa El Halabi**[*]
MIT CSAIL
marwash@mit.edu

**Slobodan Mitrović**[*]
MIT CSAIL
slobo@mit.edu

**Ashkan Norouzi-Fard**[*]
Google Zurich
ashkannorouzi@google.com

**Jakab Tardos**[*]
EPFL
jakab.tardos@epfl.ch

**Jakub Tarnawski**[*]
Microsoft Research
jatarnaw@microsoft.com

## Abstract

Submodular maximization has become established as the method of choice for the task of selecting representative and *diverse* summaries of data. However, if datapoints have sensitive attributes such as gender or age, such machine learning algorithms, left unchecked, are known to exhibit *bias*: under- or over-representation of particular groups. This has made the design of *fair* machine learning algorithms increasingly important. In this work we address the question: *Is it possible to create fair summaries for massive datasets?* To this end, we develop the first streaming approximation algorithms for submodular maximization under fairness constraints, for both monotone and non-monotone functions. We validate our findings empirically on exemplar-based clustering, movie recommendation, DPP-based summarization, and maximum coverage in social networks, showing that fairness constraints do not significantly impact utility.

## 1  Introduction

Machine learning algorithms are increasingly being used to assist human decision making. This led to concerns about the potential for bias and discrimination in automated decisions, especially in sensitive domains such as voting, hiring, criminal justice, access to credit, and higher-education [50, 20, 54, 27]. To mitigate such issues, there has been a growing effort towards developing *fair* algorithms for several fundamental problems, such as classification [59], ranking [13], clustering [16, 2, 33, 1], bandit learning [34, 46], voting [12], matching [17], influence maximization [58], and diverse data summarization [11].

In this work, we address fairness in another important class of problems, that of *streaming submodular maximization* subject to a cardinality constraint. Submodular functions are set functions that satisfy a diminishing returns property, which naturally occurs in a variety of machine learning problems. In particular, streaming submodular maximization is a natural model for *data summarization*: the task of extracting a representative subset of moderate size from a large-scale dataset. Being able to generate summaries efficiently and on-the-fly is critical to cope with the massive volume of modern datasets, which is often produced so rapidly that it cannot even be stored in memory. In many applications, such as exemplar-based clustering [23], document [44, 21] and corpus summarization [55], and recommender systems [25, 26], this challenge can be formulated as a streaming submodular maximization problem subject to a cardinality constraint. An extensive line of research focused on developing efficient algorithms in this context [14, 15, 8, 3, 51, 28].

---

[*]Equal contribution.

For monotone objectives, a one pass streaming algorithm achieving $(1/2 - \epsilon)$-approximation was proposed in [3] and shown to be tight in [29]. For non-monotone objectives, the state-of-the-art approximation is $1/5.82$, achieved by a randomized algorithm proposed in [28]. To the best of our knowledge, submodular maximization under fairness constraints has only been considered, in the *offline* setting, for monotone objectives. Celis et al. [12] provide a $(1 - 1/e)$-approximation based on the continuous greedy algorithm [9]. In this paper, we provide the first approximation algorithms for submodular maximization under fairness constraints, in the streaming setting, for both monotone and non-monotone objectives.

Characterizing what it means for an algorithm to be fair is an active area of research. Several notions of fairness have been proposed in the literature, but no universal metric of fairness exists. We adopt here the common notion used in various previous works [11, 12, 13, 17, 16], where we ask that the solution obtained is *balanced* with respect to some sensitive attribute (e.g., race, gender). Formally, we are given a set $V$ of $n$ items (e.g., people), where each item is assigned a color $c$ encoding a sensitive attribute. Let $V_1, \cdots, V_C$ be the corresponding $C$ *disjoint* groups of items sharing the same color. We say that a selection of items $S \subseteq V$ is *fair* if it satisfies $\ell_c \le |S \cap V_c| \le u_c$ for a given choice of lower and upper bounds $\ell_c, u_c \in \mathbb{Z}_{\ge 0}$, often set to be proportional to the fraction of items of color $c$, i.e., $|V_c|/n$. This definition captures several other existing notions of fairness such as statistical parity [24], diversity rules (e.g., $80\%$-rule) [19, 4], and proportional representation rules [48, 6] (see [12, Sect. 4]).

## 1.1 Our contribution

In this work, we develop a new approach for fair submodular maximization. We show how to reduce this problem to submodular maximization subject to a matroid constraint. In the case of monotone functions, our reduction preserves the approximation ratio and the number of oracle calls of the corresponding algorithm for the matroid constraint. In the non-monotone case, this reduction does not hold anymore, but it still plays an important role in our approach.

**The monotone case**  Here we achieve two results, with respect to the memory requirement. First, a $1/2$-approximate algorithm that uses an exponential in $k$ memory. This result is known to be tight due to [29]. Second, we design a low-memory efficient algorithm, matching the state-of-the-art result of the partition matroid, a special case of our problem. Namely, our proposed algorithm achieves a $1/4$-approximation using only $O(k)$ memory, and processes each element of the stream in $O(\log k)$ time and 2 oracle calls. These results are discussed in Section 4.

**The non-monotone case**  In this context, we introduce the notion of *excess ratio*, denoted by $q$ and defined as $1 - \max_{c \in C} \ell_c/|V_c|$. This refers to the "freedom" that an algorithm has in omitting elements from the solution. If the excess ratio is close to $0$, then for at least one of the colors, the total number of elements in the stream is close to the lower bound. In this case, an algorithm has little flexibility in terms of which elements it chooses from this color. Conversely, if the excess ratio is close to $1$, then the total number of elements for every color is significantly higher than the corresponding lower bound.

We show that the excess ratio is closely tied to the hardness of fair non-monotone submodular maximization in the streaming setting. Indeed, we propose a $q/5.82$-approximation algorithm using $O(k)$ memory, and then show that any algorithm that achieves a better than $q$-approximation requires $\Omega(n)$ memory. These results are discussed in Section 5. Note that in practice, the size of the summary is expected to be significantly smaller than the size of the input. Hence, it is natural to expect that the excess ratio will be close to $1$, and thus our algorithm will perform well on real-world applications.

**Empirical evaluation**  We study the empirical performance of our algorithms on various real-life tasks where being fair is important. We observe that our algorithms allow us to enforce fairness constraints at the cost of a small loss in utility, while also matching the efficiency and number of oracle calls of "unfair" state-of-the-art algorithms.

## 1.2 Additional related work

Submodular maximization has been extensively studied. The setting most similar to ours is that of streaming submodular maximization under a matroid constraint. The first result in this setting, for

monotone functions, is by [14] which proposed a $1/4p$-approximation algorithm under $p$-matroid constraints using $O(k)$ memory, which was later extended to $p$-matchoid constraints in [15]. For a single matroid constraint, the best known approximation is achieved by [32], who proposed a $1/2$-approximation algorithm using $k^{O(k)}$ memory. This is essentially tight, as [29] shows that a $(1/2 + \epsilon)$-approximation of monotone submodular maximization requires $\Omega(n)$ space, even for cardinality constraint, for any positive $\epsilon$. For non-monotone functions, the first streaming algorithm for this problem appears in [15], which achieves an approximation ratio of $(1-\epsilon)(2-o(1))/(8+e)p$ with $O(k \log k)$ memory. This was improved in [28] to a $1/(2p + 2\sqrt{p(p+1)} + 1)$-approximation using $O(k)$ memory. The latter implies the best known result for non-monotone functions under a single matroid constraint, with $1/(3 + 2\sqrt{2}) \approx 1/5.82$-approximation.

In the sequential setting, [12] studied the fair multiwinner voting problem, which they cast as a fair submodular maximization problem, and presented a $(1 - 1/e)$-approximation algorithm for it. They also considered the setting in which color groups can overlap. In this setup, even checking feasibility is NP-hard, when elements can belong to 3 or more colors. Nevertheless, if fairness constraints are allowed to be *nearly* satisfied, [12] gives a $(1 - 1/e - o(1))$-approximation algorithm. [36] studied data summarization with privacy and fairness constraints, but adopted a different notion of fairness, where part of the data is deleted or masked due to fairness criteria.

## 2 Preliminaries

We consider a (potentially large) collection $V$ of $n$ items, also called the *ground set*. We study the problem of maximizing a *non-negative submodular function* $f : 2^V \to \mathbb{R}_{\geq 0}$. Given two sets $X, Y \subseteq V$, the *marginal gain* of $X$ with respect to $Y$ is defined as

$$f(X \mid Y) = f(X \cup Y) - f(Y),$$

which quantifies the change in value when adding $X$ to $Y$. The function $f$ is *submodular* if for any two sets $X$ and $Y$ such that $X \subseteq Y \subseteq V$ and any element $e \in V \setminus Y$ we have

$$f(e \mid X) \geq f(e \mid Y).$$

We say that $f$ is *monotone* if for any element $e \in V$ and any set $Y \subseteq V$ it holds that $f(e \mid Y) \geq 0$; otherwise, if $f(e \mid Y) < 0$ for some $e \in V$ and $Y \subseteq V$, we say that $f$ is *non-monotone*. Throughout the paper, we assume that $f$ is given in terms of a value oracle that computes $f(S)$ for given $S \subseteq V$. We also assume that $f$ is *normalized*, i.e., $f(\emptyset) = 0$.

**Fair submodular maximization**   We assume that the ground set $V$ is colored so that each element has exactly one color. We index the colors $c = 1, 2, ..., C$ and denote by $V_c$ the set of elements of color $c$. Thus $V = V_1 \cup ... \cup V_C$ is a partition. For each color $c$ we assume that we are given a lower and an upper bound on the number of elements of color $c$ that a feasible solution must contain. These represent fairness constraints and are denoted by $\ell_c$ and $u_c$, respectively. Let $k \in \mathbb{Z}_{\geq 0}$ be a global cardinality constraint. We denote by $\mathcal{F}$ the set of solutions feasible under these fairness and cardinality constraints, i.e.,

$$\mathcal{F} = \{S \subseteq V : |S| \leq k, |S \cap V_c| \in [\ell_c, u_c] \text{ for all } c = 1, ..., C\}.$$

The problem of maximizing a function $f$ under *cardinality and fairness constraints* is defined as selecting a set $S \subseteq V$ with $S \in \mathcal{F}$ so as to maximize $f(S)$. We use OPT to refer to a set maximizing $f$. We assume that there exists a feasible solution, i.e., $\mathcal{F} \neq \emptyset$. In particular, this implies that $\sum_{c=1}^{C} \ell_c \leq k$.

**Matroids**   In our algorithms we often reduce to submodular maximization under a matroid constraint: the problem of selecting a set $S \subseteq V$ with $S \in \mathcal{M}$ so as to maximize $f(S)$, where $\mathcal{M}$ is a matroid. We call a family of sets $\mathcal{M} \subseteq 2^V$ a *matroid* if it satisfies the following properties: $\mathcal{M} \neq \emptyset$; *downward-closedness:* if $A \subseteq B$ and $B \in \mathcal{M}$, then $A \in \mathcal{M}$; *augmentation:* if $A, B \in \mathcal{M}$ with $|A| < |B|$, then there exists $e \in B$ such that $A + e \in \mathcal{M}$.

## 3 Warm-up: Monotone Sequential Algorithm

In this section, we consider the classic sequential setting and assume that the submodular function $f$ is monotone. We present a natural greedy algorithm FAIR-GREEDY, and show that it achieves

a $1/2$-approximate solution. The advantage of this algorithm compared to the algorithm provided in [12] based on continuous greedy, is its simplicity and faster running time of $O(|V|k)$. Moreover, the algorithm and ideas introduced in this section serve as a warm-up for the streaming setting.

The greedy algorithm picks at each step the element that has the largest marginal gain while satisfying some constraint. We start by observing that if this element was only required to satisfy the upper-bound and cardinality constraints, the greedy algorithm might not return a feasible solution. It might reach the global cardinality constraint without satisfying the lower bounds. Therefore, a more careful selection of the elements is needed. To that end, we define the following concept.

**Definition 3.1** *We call a set $S \subseteq V$ extendable if it is a subset $S \subseteq S'$ of some feasible solution $S' \in \mathcal{F}$.*

For a set $S$ to be extendable, it must satisfy the upper bounds: $|S| \leq k$ and $|S \cap V_c| \leq u_c$ for all $c = 1, ..., C$. If $S$ also satisfies the lower bounds ($|S \cap V_c| \geq \ell_c$ for all $c$), then $S$ is already feasible. Otherwise, it is necessary to add at least $\ell_c - |S \cap V_c|$ elements of every color $c$ for which $S$ does not yet satisfy the lower bound. This yields a feasible extension as long as it does not violate the global cardinality constraint $k$. In short, we have the following simple characterization:

**Observation 3.2** *A set $S \subseteq V$ is extendable if and only if*

$$|S \cap V_c| \leq u_c \quad \text{for all } c = 1, ..., C \qquad \text{and} \qquad \sum_{c=1}^{C} \max(|S \cap V_c|, \ell_c) \leq k \,.$$

The FAIR-GREEDY algorithm starts with $S = \emptyset$ and in each step takes the element with highest marginal gain that keeps the solution extendable.

**Fact 3.3** FAIR-GREEDY *is a $1/2$-approximate algorithm with $O(|V|k)$ running time for fair monotone submodular maximization.*

The analysis of FAIR-GREEDY is deferred to Appendix A.

---

**Algorithm 1** FAIR-GREEDY

1: $S \leftarrow \emptyset$
2: **while** $|S| < k$ **do**
3: $\quad U \leftarrow \{e \in V \mid S + e \text{ is extendable}\}$
4: $\quad S \leftarrow S + \text{argmax}_{e \in U} f(e \mid S)$
5: **return** $S$

---

## 4 Monotone Streaming Algorithm

In this section, we present our algorithm for fair *monotone* submodular maximization in the streaming setting, and we prove its approximation guarantee. We begin by explaining the intuition behind our algorithm. If we removed the lower-bound constraints $|S \cap V_c| \geq \ell_c$ in $\mathcal{F}$, then the remaining constraints would give rise to a matroid (a so-called laminar matroid). There exist efficient streaming algorithms for submodular maximization under matroid constraint (e.g. [14, 28]), which we could use in a black-box manner. A solution obtained from such an algorithm $\mathcal{A}$ may of course violate the lower-bound constraints. We could hope to augment our solution to a feasible one using "backup" elements gathered from the stream in parallel to $\mathcal{A}$. As we are dealing with a *monotone* submodular function, adding such elements would not hurt the approximation guarantee inherited from $\mathcal{A}$.

However, doing so might violate the global cardinality constraint $|S| \leq k$. Indeed, as we remarked in Section 3, not every set satisfying the upper-bound constraints can be extended to a feasible solution. Recall that the right constraint to place was for the solution to be *extendable* (Definition 3.1) to a feasible set. Crucially, we show that such a solution can be efficiently found, as extendable subsets of $V$ form a matroid.

**Lemma 4.1** *Let $\tilde{\mathcal{F}} \subseteq 2^V$ be the family of all extendable subsets of $V$. Then $\tilde{\mathcal{F}}$ is a matroid.*

The proof of Lemma 4.1 can be found in Appendix B.1. Algorithms for submodular maximization under a matroid constraint require access to a membership oracle for the matroid. For $\tilde{\mathcal{F}}$, membership is easy to verify, as follows from Observation 3.2.

Now we are ready to present our algorithm FAIR-STREAMING for fair monotone submodular maximization. Let $\mathcal{A}$ be a streaming algorithm for monotone submodular maximization under a matroid constraint. FAIR-STREAMING runs $\mathcal{A}$ to construct an extendable set $S_{\mathcal{A}}$ that approximately maximizes $f$. In parallel, for every color $c$ we collect a backup set $B_c$ of size $|B_c| = \ell_c$. At the end, the solution $S_{\mathcal{A}}$ is augmented to a feasible solution $S$ using a simple

---

**Algorithm 2** FAIR-STREAMING

1: $S_{\mathcal{A}} \leftarrow \emptyset, B_c \leftarrow \emptyset$ for all $c = 1, ..., C$
2: **for** every arriving element $e$ of color $c$ **do**
3:     process $e$ with algorithm $\mathcal{A}$
4:     **if** $|B_c| < \ell_c$ **then**
5:         $B_c \leftarrow B_c + e$
6: $S_{\mathcal{A}} \leftarrow$ solution of algorithm $\mathcal{A}$
7: $S \leftarrow S_{\mathcal{A}}$ augmented with elements in sets $B_c$
8: **return** $S$

---

procedure: for every color such that $|S_{\mathcal{A}} \cap V_c| < \ell_c$, add any $\ell_c - |S_{\mathcal{A}} \cap V_c|$ elements from $B_c$ to satisfy the lower bound. The pseudocode of FAIR-STREAMING is given as Algorithm 2. Thus we get the following black-box reduction, proved in Appendix B.2.

**Theorem 4.2** *Suppose $\mathcal{A}$ is a streaming algorithm for monotone submodular maximization under a matroid constraint. Then there exists a streaming algorithm for fair monotone submodular maximization with the same approximation ratio and memory usage as $\mathcal{A}$.*

Applying Theorem 4.2 to the algorithm of [32] we get the following result.

**Theorem 4.3 (Streaming monotone)** *There exists a streaming algorithm for fair monotone submodular maximization that attains $1/2$-approximation and uses $k^{O(k)}$ memory.*

We remark that the $1/2$ approximation ratio is tight even in the simpler setting of monotone streaming submodular maximization subject to a cardinality constraint [29].
A more practical algorithm to use as $\mathcal{A}$ in FAIR-STREAMING is the $1/4$-approximation algorithm of Chakrabarti and Kale [14]. It turns out that we can further adapt and optimize our implementation to make our algorithm extremely efficient and use only 2 oracle calls and $O(\log k)$ time per element. We prove Theorem 4.4 in Appendix C, where we also state the algorithm of [14] for completeness.

**Theorem 4.4 (Streaming monotone)** *There exists a streaming algorithm for fair monotone submodular maximization that attains $1/4$-approximation, using $O(k)$ memory. This algorithm uses $O(\log k)$ time and 2 oracle calls per element.*

## 5 Non-monotone Streaming Case

We now focus on non-monotone functions. One might consider applying the approach from the previous section, i.e., use a known algorithm for non-monotone submodular maximization under a matroid constraint to find a high quality extendable solution, and then add backup elements to satisfy the lower-bound constraints. However, this approach is more challenging now, as adding backup elements to a solution could drastically decrease its value.

For example, consider the following instance with two colors. Let $V = A \cup B \cup \{x\}$ where $A = \{a_i | i \in [m_1]\}$, $B = \{b_i | i \in [m_2]\}$, each $e \in A \cup B$ is *blue*, and $x$ is *red*. Let $f(S) = |S|$ for each $S \subseteq A \cup B$, and let $x$ "nullify" the contributions of $B$ but not the contributions of $A$. Formally,

$$f(S) = \begin{cases} |S| & \text{if } x \notin S, \\ |S \cap A| & \text{if } x \in S. \end{cases}$$

It is easy to verify that $f$ is submodular (a formal proof is given in the Appendix). Suppose that we have to pick exactly one red element, i.e., $\ell_{\text{red}} = u_{\text{red}} = 1$. This renders all elements in $B$ useless, and the optimal solution takes only elements in $A$. However, before $x$ appears, elements in $A$ and $B$ are *indistinguishable*, since $f(S) = |S|$ for any $S \subseteq A \cup B$. Therefore, if $m_1 \ll m_2$, and $x$ is last in the stream, any algorithm that does not store the entirety of $V$ will pick only a few elements from $A$, thus achieving almost zero objective value once $x$ is included in the solution.

The core difficulty here, and in general, is that $\ell_c$ is nearly as large as $n_c = |V_c|$ for some color $c$, like for red in our example. To quantify this we introduce the *excess ratio*

$$q = 1 - \max_{c \in [C]} \ell_c / n_c.$$

We show that this quantity is inherent to the difficulty of the problem. Indeed, it is impossible to achieve an approximation ratio better that $q$ with sublinear space.

**Theorem 5.1 (Hardness non-monotone)** *For any constant $\epsilon > 0$ and $q \in [0, 1]$, any algorithm for fair non-monotone submodular maximization that outputs a $(q + \epsilon)$-approximation for inputs with excess ratio above $q$, with probability at least $2/3$, requires $\Omega(n)$ memory.*

The proof of Theorem 5.1 is deferred to Appendix D.2. Note that in practice $q$ is nearly always large, as the size of the data is significantly larger than the size of the summary. In what follows, we present a streaming algorithm for fair non-monotone submodular maximization, that nearly matches the above approximation lower-bound, using only $O(k)$ memory.

## 5.1 Non-monotone algorithm

Our non-monotone algorithm FAIR-SAMPLE-STREAMING is a variant of FAIR-STREAMING, where we modify the way backup elements are collected. Let $\mathcal{A}$ be an $\alpha$-approximation algorithm for non-monotone submodular maximization under a matroid constraint. FAIR-SAMPLE-STREAMING runs algorithm $\mathcal{A}$ to construct an extendable set $S_{\mathcal{A}}$ that approximately maximizes $f$. In parallel, our algorithm collects for every color $c$ a backup set $B_c$ of size

| **Algorithm 3** FAIR-SAMPLE-STREAMING |
| --- |
| 1: $S_{\mathcal{A}} \leftarrow \emptyset, B_c \leftarrow \emptyset$ for all $c = 1, ..., C$ |
| 2: **for** every arriving element $e$ **do** |
| 3:     process $e$ with algorithm $\mathcal{A}$ |
| 4:     **if** $e \in V_c$ **then** |
| 5:         $B_c \leftarrow \text{Reservoir-Sample}(B_c, e)$ |
| 6: $S \leftarrow S_{\mathcal{A}}$ augmented with elements in sets $B_c$ |
| 7: **return** $S$ |

$|B_c| = \ell_c$, by sampling without replacement $\ell_c$ elements in $V_c$ using reservoir sampling [43]. Note that we do not need to know the value of $n_c$ to execute reservoir sampling. At the end, the solution $S_{\mathcal{A}}$ is augmented to a feasible solution $S$ using the same simple procedure as in Section 4. The pseudocode of FAIR-SAMPLE-STREAMING is given as Algorithm 3. We show that adding elements from the back-up set reduces the objective value by a factor of at most $q$.

**Theorem 5.2** *Suppose $\mathcal{A}$ is a streaming $\alpha$-approximate algorithm for non-monotone submodular maximization under a matroid constraint. Then, there exists a streaming algorithm for fair non-monotone submodular maximization with expected $q\alpha$ approximation ratio, and the same memory usage, oracle calls, and running time as $\mathcal{A}$.*

The proof is provided in Appendix D.1. Combining this with the state of the art $1/5.82$-approximation algorithm of Feldman, et al. [28] (restated in Appendix C for completeness) yields the following.

**Theorem 5.3 (Streaming non-monotone)** *There exists a streaming algorithm for fair non-monotone submodular maximization that achieves $q/5.82$-approximation in expectation, using $O(k)$ memory. This algorithm uses $O(k)$ time and $O(k)$ oracle calls per element.*

## 6 Empirical Evaluation

In this section, we empirically validate our results and address the question: *What is the price of fairness?* To this end, we compare our approach against several baselines on four datasets. We measure: (1) Objective values. (2) Violation of fairness constraints: Given a set $S$, we define $\text{err}(S) = \sum_{c \in [C]} \max\{|S \cap V_c| - u_c, \ell_c - |S \cap V_c|, 0\}$. A single term in this sum quantifies by how many elements $S$ violates the lower or upper bound. Note that $\text{err}(S)$ is in the range $[0, 2k]$. (3) Number of oracle calls, as is standard in the field to measure the efficiency of algorithms.

We compare the following algorithms:

- **FAIR-STREAMING-CK**: monotone, $\mathcal{A} = $ Chakrabarti-Kale [14] (Theorem 4.4 and Appendix C.1),
- **FAIR-STREAMING-FKK**: monotone, $\mathcal{A} = $ Feldman et al. [28] (Appendix C.2),
- **FAIR-SAMPLE-STREAMING-FKK**: non-monotone, $\mathcal{A} = $ Feldman et al. [28] (Theorem 5.3),

- **UPPERBOUNDS**: [28] (Appendix C.2) applied to matroid defining upper bounds only ($u_c$ and $k$),
- **FAIR-GREEDY**: monotone; where data size allows; see Section 3,
- **GREEDY**: monotone; when data size allows; no fairness constraints, only $k$,
- **SIEVESTREAMING**: Badanidiyuru et al. [3]; monotone; no fairness constraints, only $k$,
- **RANDOM**: maintain random sample of $k$ elements; no fairness constraints,
- **FAIR-RANDOM**: maintain random feasible (fair) solution.

We now describe our experiments. We report the results in Fig. 1, and discuss them in Section 6.5. The code is available at `https://github.com/google-research/google-research/tree/master/fair_submodular_maximization_2020`.

## 6.1 Maximum coverage

Social influence maximization [37] and network marketing [42] are some of the prominent applications of the *maximum coverage* problem. The goal of this problem is to select a fixed number of nodes that maximize the coverage of a given network. Given a graph $G = (V, E)$, let $N(v) = \{u : (v, u) \in E\}$ denote the neighbors of $v$. Then the coverage of $S \subseteq V$, denoted by $f(S)$, is defined as the monotone submodular function $f(S) = \left| \bigcup_{v \in S} N(v) \right|$. We perform experiments on the Pokec social network [41]. This network consists of $1\,632\,803$ nodes, representing users, and $30\,622\,564$ edges, representing friendships. Each user profile contains attributes such as age, height and weight; these can take value "null". We impose fairness constraints with respect to (i) age and (ii) body mass index (BMI).

(i) We split ages into ranges $[1, 10], [11, 17], [18, 25], [26, 35], [36, 45], [46+]$ and consider each range as one color. We create another color for records with "null" age (around $30\%$). Then for every color $c$ we set $\ell_c = \max\{0, |V_c|/n - 0.05\} \cdot k$ and $u_c = \min\{1, |V_c|/n + 0.05\} \cdot k$, except for the null color, where we set $\ell_c = 0$. The results are shown in Fig. 1a, 1b, and 1c.

(ii) BMI is computed as the ratio between weight (in kg) and height (in m) squared. Around $60\%$ of profiles do not have set height or weight. We discard all such profiles, as well as profiles with clearly fake data (less than $2\%$ of profiles). The resulting graph consists of $582\,319$ nodes and $5\,834\,695$ edges. The profiles are colored with respect to four standard BMI categories (underweight, normal weight, overweight and obese). Lower- and upper-bound fairness constraints are set again to be within $5\%$ of their respective frequencies. The results are shown in Fig. 1d, 1e, and 1f.

## 6.2 Movie recommendation

We use the Movielens 1M dataset [31], which contains $\sim$1M ratings for $3\,900$ movies by $6\,040$ users, to develop a movie recommendation system. We follow the experimental setup of prior work [47, 51]: we compute a low-rank completion of the user-movie rating matrix [57], which gives rise to feature vectors $w_u \in \mathbb{R}^{20}$ for each user $u$ and $v_m \in \mathbb{R}^{20}$ for each movie $m$. Then $w_u^\top v_m$ approximates the rating of $m$ by $u$. The (monotone submodular) utility function for a collection $S$ of movies personalized for user $u$ is defined as

$$f_u(S) = \alpha \cdot \sum_{m' \in M} \max \left( \max_{m \in S} \left( v_m^\top v_{m'} \right), 0 \right) + (1 - \alpha) \cdot \sum_{m \in S} w_u^\top v_m.$$

The first term optimizes coverage of the space of all movies (enhancing diversity) [45], and the second term sums up user-dependent movie scores; $\alpha$ controls the trade-off between the two terms. In our experiment we recommend a collection of movies for $\alpha = 0.85$ and $k$-values up to 100. Each movie in the database is assigned to one of 18 genres $c$; our fairness constraints mandate a representation of genres in $S$ similar to that in the entire dataset. More precisely, we set $\ell_c = \left\lfloor 0.8 \frac{|V_c|}{|V|} k \right\rfloor$ and $u_c = \left\lceil 1.4 \frac{|V_c|}{|V|} k \right\rceil$. The results are shown in Fig. 1g, 1h, and 1i.

## 6.3 Census DPP-based summarization

A common reliable method for data summarization is to use a Determinantal Point Process (DPP) to assign a diversity score to each subset, and choose the subset that maximizes this score. A DPP is

a probability measure over subsets, defined for every $S \subseteq V$, as $P(S) = \frac{\det(L_S)}{\det(I+L)}$, where $L$ is an $n \times n$ positive semi-definite kernel matrix, $L_S$ is the $|S| \times |S|$ principal submatrix of $L$ indexed by $S$, and $I$ is the identity matrix. To find the most diverse representative subset, we need to maximize the non-montone submodular function $f(S) = \log \det(L_S)$ [40]. To ensure non-negativity (on non-empty sets), we normalize $f$ by a constant.

We use the Census Income dataset [22] which consists of $45\,222$ records extracted from the 1994 Census database, with 14 attributes such as age, race, gender, education, and whether the income is above or below 50K USD. We follow the experimental setup of [11] to generate feature vectors of dimension 992 for 5000 randomly chosen records.[2] We select fair representative summaries with respect to race, requiring that each of the race categories provided in the dataset (White, Black, Asian-Pac-Islander, Amer-Indian-Eskimo, and Other) have a similar representation in $S$ as in the entire dataset. Accordingly, we set $\ell_c = \left\lfloor 0.9 \frac{T}{n} k \right\rfloor$ and $u_c = \left\lceil 1.1 \frac{T}{n} k \right\rceil$, and vary $k$ between $50 - 600$. The results are shown in Fig. 1j, 1k, and 1l.

## 6.4 Exemplar-based clustering

We consider a dataset containing one record for each phone call in a marketing campaign ran by a Portuguese banking institution [49]. We aim to find a representative subset of calls in order to assess the quality of service. We choose numeric attributes such as client age, gender, account balance, call date, and duration, to represent each record in the Euclidean space. We require the chosen subset to have clients in a wide range of ages. We divide the records into six groups according to age: $[0, 29], [30, 39], [40, 49], [50, 59], [60, 69], [70+]$; the numbers of records in each range are respectively: $5\,273, 18\,089, 11\,655, 8\,410, 1\,230, 554$. We set our bounds so as to ensure that each group comprises $10 - 20\%$ of the subset. Then we maximize the following monotone submodular function [38], where $R$ denotes all records:

$$f(S) = C - \sum_{r \in R} \min_{e \in S} d(r, e) \quad \text{where} \quad d(x, y) = \|x - y\|_2^2 .$$

We let $f(\emptyset) = 0$ and $C$ be $|V|$ times the maximum distance.[3] The results are shown in Fig. 1m, 1n, and 1o, where the clustering cost refers to $C - f(S)$.

## 6.5 Results

We observe that in all the experiments our algorithms make smaller or similar number of oracle calls compared to the baselines in corresponding settings (streaming or sequential). Moreover, the objective value of the fair solutions obtained by our algorithms is similar to the unfair baseline solutions, with less than $15\%$ difference.

We also observe that the algorithms that do not impose fairness constraints introduce significant bias. For example, SIEVESTREAMING makes 150 errors in the maximum coverage experiment for $k = 200$ (see Fig. 1b), and 30 errors for $k = 70$ in the exemplar-based clustering experiment (see Fig. 1n). Moreover, even though UPPERBOUNDS satisfies the upper-bounds constraints, it still makes a noticeable amount of errors. For instance, it makes 20 errors in the maximum coverage experiment for $k = 200$ (see Fig. 1b), and 100 errors in the DPP-based summarization experiment for $k = 600$ (see Fig. 1k).

## 7 Conclusion

We presented the first streaming approximation algorithms for fair submodular maximization, for both monotone and non-monotone objectives. Our algorithms efficiently generate balanced solutions with respect to a sensitive attribute, while using asymptotically optimal memory. We empirically demonstrate that fair solutions are often nearly optimal, and that explicitly imposing fairness constraints is necessary to ensure balanced solutions.

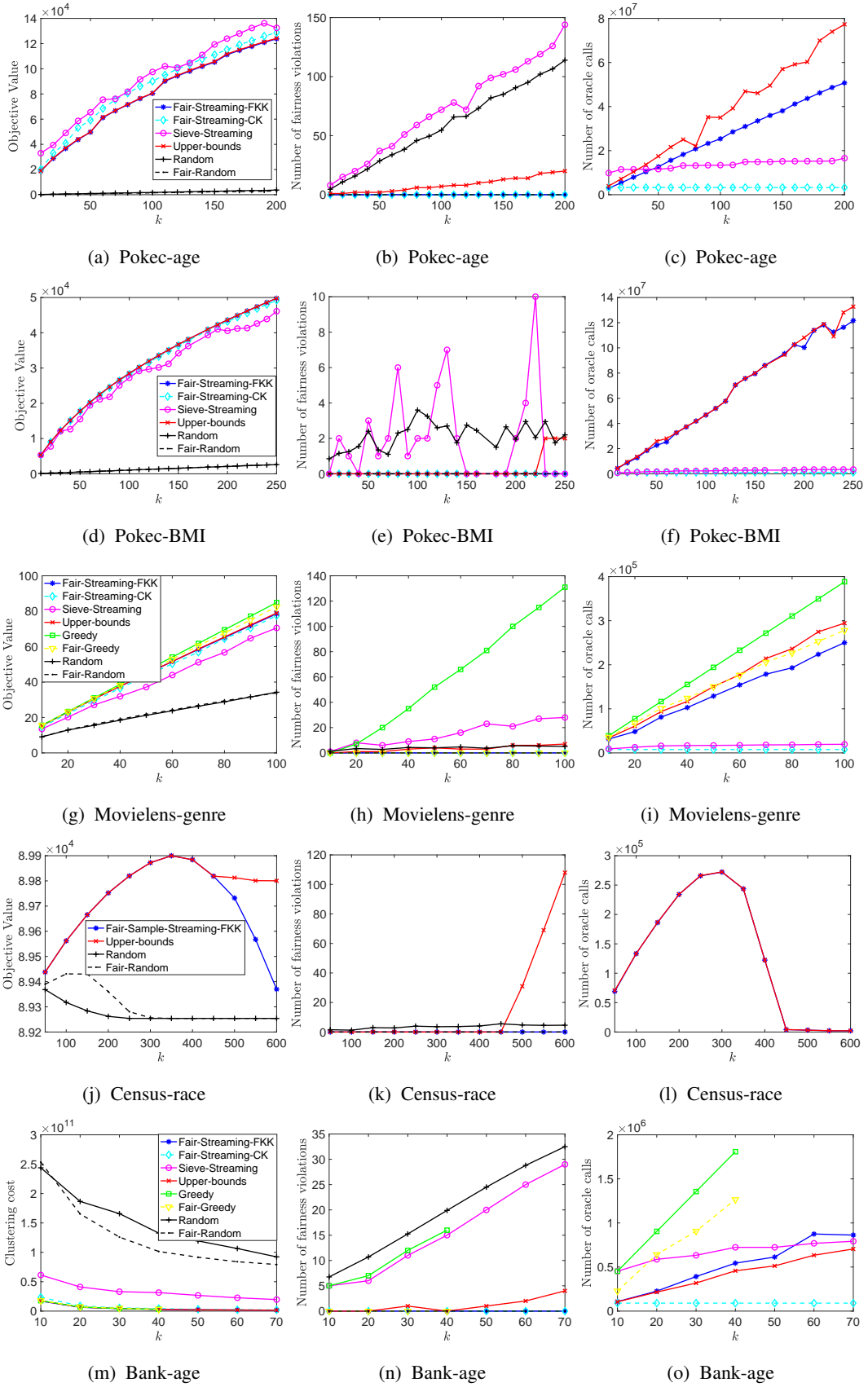

Figure 1: Performance of FAIR-STREAMING-CK, FAIR-STREAMING-FKK and FAIR-SAMPLE-STREAMING-FKK compared to other baselines, in terms of objective value, violation of fairness constraints, and running time, on Movielens, Pokec, Census, and Bank-Marketing datasets.

## Broader Impact

Several recent studies have shown that automated data-driven methods can unintentionally lead to bias and discrimination [35, 56, 5, 10, 52]. Our proposed algorithms will help guard against these issues in data summarization tasks arising in various settings – from electing a parliament, over selecting individuals to influence for an outreach program, to selecting content in search engines and news feeds. As expected, fairness does come at the cost of a small loss in utility value, as observed in Section 6. It is worth noting that this "price of fairness" (i.e., the decrease in optimal objective value when fairness constraints are added) should not be interpreted as fairness leading to a less desirable outcome, but rather as a trade-off between two valuable metrics: the original application-dependent utility, and the fairness utility. Our algorithms ensure solutions achieving a close to optimal trade-off.

Finally, despite the generality of the fairness notion we consider, it does not capture certain other notions of fairness considered in the literature (see e.g., [18, 58]). No universal metric of fairness exists. The question of which fairness notion to employ is an active area of research, and will be application dependent.

## Acknowledgments and Disclosure of Funding

Marwa El Halabi was supported by a DARPA D3M award, NSF CAREER award 1553284, NSF award 1717610, and by an ONR MURI award. The views, opinions, and/or findings contained in this article are those of the authors and should not be interpreted as representing the official views or policies, either expressed or implied, of the Defense Advanced Research Projects Agency or the Department of Defense. Slobodan Mitrović was supported by the Swiss NSF grant No. P400P2_191122/1, MIT-IBM Watson AI Lab and Research Collaboration Agreement No. W1771646, and FinTech@CSAIL. Jakab Tardos has received funding from the European Research Council (ERC) under the European Union's Horizon 2020 research and innovation programme (grant agreement No 759471).

## Footnotes

[2]Code available at: `https://github.com/DamianStraszak/FairDiverseDPPSampling`.

[3]Note that $C$ is added to ensure that all values are non-negative. Any $C$ with this property would be suitable.

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
