[Supplementary Material]

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

(a) Pokec-age     (b) Pokec-age     (c) Pokec-age

(d) Pokec-BMI     (e) Pokec-BMI     (f) Pokec-BMI

(g) Movielens-genre     (h) Movielens-genre     (i) Movielens-genre

(j) Census-race     (k) Census-race     (l) Census-race

(m) Bank-age     (n) Bank-age     (o) Bank-age

Figure 1: Performance of FAIR-STREAMING-CK, FAIR-STREAMING-FKK and FAIR-SAMPLE-STREAMING-FKK compared to other baselines, in terms of objective value, violation of fairness constraints, and running time, on Movielens, Pokec, Census, and Bank-Marketing datasets.

## Broader Impact

Several recent studies have shown that automated data-driven methods can unintentionally lead to bias and discrimination [35, 56, 5, 10, 52]. Our proposed algorithms will help guard against these issues in data summarization tasks arising in various settings – from electing a parliament, over selecting individuals to influence for an outreach program, to selecting content in search engines and news feeds. As expected, fairness does come at the cost of a small loss in utility value, as observed in Section 6. It is worth noting that this "price of fairness" (i.e., the decrease in optimal objective value when fairness constraints are added) should not be interpreted as fairness leading to a less desirable outcome, but rather as a trade-off between two valuable metrics: the original application-dependent utility, and the fairness utility. Our algorithms ensure solutions achieving a close to optimal trade-off.

Finally, despite the generality of the fairness notion we consider, it does not capture certain other notions of fairness considered in the literature (see e.g., [18, 58]). No universal metric of fairness exists. The question of which fairness notion to employ is an active area of research, and will be application dependent.

## Acknowledgments and Disclosure of Funding

Marwa El Halabi was supported by a DARPA D3M award, NSF CAREER award 1553284, NSF award 1717610, and by an ONR MURI award. The views, opinions, and/or findings contained in this article are those of the authors and should not be interpreted as representing the official views or policies, either expressed or implied, of the Defense Advanced Research Projects Agency or the Department of Defense. Slobodan Mitrović was supported by the Swiss NSF grant No. P400P2_191122/1, MIT-IBM Watson AI Lab and Research Collaboration Agreement No. W1771646, and FinTech@CSAIL. Jakab Tardos has received funding from the European Research Council (ERC) under the European Union's Horizon 2020 research and innovation programme (grant agreement No 759471).

## Footnotes

[2]Code available at: `https://github.com/DamianStraszak/FairDiverseDPPSampling`.

[3]Note that $C$ is added to ensure that all values are non-negative. Any $C$ with this property would be suitable.

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

## A  Details of FAIR-GREEDY

### A.1  Proof of Fact 3.3

We remark that, once we know that extendable sets form a matroid (Lemma 4.1), the approximation ratio of FAIR-GREEDY can be seen to follow from the fact that the greedy algorithm for submodular maximization under matroid constraints achieves a $1/2$-approximation guarantee [30]. For completeness, below we also give a self-contained proof.

*Proof.* Let an optimal solution be $O \subseteq V$ and let the output of greedy be $G = \{g_1, \ldots, g_k\}$, where the elements where chosen in the order $g_1, \ldots g_k$ by the algorithm. To prove the lemma, we will show that $f(G) \geq \frac{1}{2} \cdot f(G \cup O)$.

Let us order the elements of $O$ as $o_1, \ldots, o_k$ in a way that the colors of $g_j$ and $o_j$ coincide as much as possible. Specifically, we want an ordering such that for all $j$

- either $g_j$ and $o_j$ are the same color
- or, if $c(g_j) = c_1$ and $c(o_j) = c_2$ are different, then $|G \cap V_{c_1}| > |O \cap V_{c_1}|$ and $|G \cap V_{c_2}| < |O \cap V_{c_2}|$,

where $c(v)$ denotes the color of element $v \in V$. Such a matching between elements of $G$ and $O$ can be easily constructed recursively. Indeed, as long as there remain elements of $G$ and $O$ that are the same color match them together; once all remaining elements are of different color match them arbitrarily.

**Claim A.1** *For any $j$, $G \backslash \{g_j\} \cup \{o_j\}$ is a feasible solution.*

Indeed, if $g_j$ and $o_j$ are the same color, exchanging them does not change the color profile of $G$ and it remains feasible. On the other hand, if $c(g_j) = c_1$ and $c(o_j) = c_2$ are different, then $|G \cap V_{c_1}| > |O \cap V_{c_1}| \geq \ell_{c_1}$, and removing $g_j$ from $G$ does not violate any conditions. Similarly, $|G \cap V_{c_2}| < |O \cap V_{c_2}| \geq u_{c_2}$ and adding $o_j$ to $G$ does not violate any conditions either.

From Claim A.1 it follows that $g_1, \ldots, g_{j-1}, o_j$ is a feasible partial solution (see Definition 3.1). Therefore, by the definition of FAIRGREEDY, $f(g_j | g_1, \ldots, g_{j-1}) \geq f(o_j | g_1, \ldots, g_{j-1})$.

Therefore,

$$
\begin{aligned}
f(G) &= \sum_{j=1}^{k} f(g_j | g_1, \ldots, g_{j-1}) \\
&\geq \sum_{j=1}^{k} f(o_j | g_1, \ldots, g_{j-1}) \\
&\geq \sum_{j=1}^{k} f(o_j | g_1, \ldots, g_k, o_1, \ldots, o_{j-1}) \\
&= f(O | G),
\end{aligned}
$$

and so

$$
2f(G) \geq f(O \cup G) \geq f(O).
$$

$\square$

## A.2 Checking extendability

In order for Algorithm 1 to run in time $O(|V|k)$ we must solve the problem of generating the set $U = \{e \in V \mid S + e \text{ is extendable}\}$ in $O(|V|)$ time. That is, we must be able to check if adding a single element $e$ to our set $S$ would maintain extendability in $O(1)$ time.

This can be done by simply maintaining the counts $t_c = |S \cap V_c|$ of elements of each color in $S$, as well as the sum $Q = \sum_{c=1}^{C} \max(t_c, \ell_c)$. Recall Observation 3.2 which states that $S$ is extendable if $t_c \leq u_c$ for each $c$ and $Q \leq k$.

At the beginning of our algorithm we initialize these variables. Then, whenever a potential extension $e$ of color $c$ is considered, we call CANDIDATE$(c)$ to determine whether adding it would maintain extendability. Once we augment $S$ with an element $e$ of color $c$, we update the stored variables using UPDATE$(c)$.

---

**Algorithm 4** Checking extendability

> **procedure** INITIALIZE
>> **for** $c \in [C]$ **do**
>>> $t_c \leftarrow 0$
>>
>> $Q \leftarrow \sum_{c=1}^{C} \ell_c$
>
> **procedure** UPDATE$(c)$
>> $t_c \leftarrow t_c + 1$
>> **if** $t_c > \ell_c$ **then**
>>> $Q \leftarrow Q + 1$
>
> **procedure** CANDIDATE$(c)$
>> **if** $t_c = u_c$ **then**
>>> **return false**
>>
>> **if** $t_c < \ell_c$ **then**
>>> **return true**
>>
>> **if** $\ell_c \leq t_c < u_c$ **then**
>>> **return** $Q < k$

---

To implement the non-monotone submodular maximization algorithm of [28] which we use in Section 5.1, it is also useful to be able to verify whether a pair of elements can be swapped in the current solution. Suppose we are trying to add element $e_1$ of color $c_1$ to $S$, while removing element $e_2$ of color $c_2$. To verify if this is legal, we call SWAP$(c_1, c_2)$.

---

**Algorithm 5** Checking extendability

> **procedure** SWAP$(c_1, c_2)$
>> **if** $c_1 = c_2$ **then**
>>> **return true**
>>
>> **if** $t_{c_1} = u_{c_1}$ **then**
>>> **return false**
>>
>> **if** $Q = k$ **and** $t_{c_1} \geq \ell_{c_1}$ **and** $t_{c_2} \leq \ell_{c_2}$ **then**
>>> **return false**
>>
>> **else**
>>> **return true**

---

# B Monotone Streaming – Proofs

## B.1 Proof of Lemma 4.1

*Proof.* Let $\mathcal{B}$ consist of all maximal sets in $\mathcal{F}$. We will show that $\mathcal{B}$ satisfies the following two axioms.

(B1) $\mathcal{B} \neq \emptyset$.

(B2) If $B_1, B_2 \in \mathcal{B}$ and $x \in B_1 \setminus B_2$, then there exists $y \in B_2 \setminus B_1$ such that $B_1 - x + y \in \mathcal{B}$.

These axioms imply (see e.g. [53, Theorem 1.2.3]) that the downward closure (collection of all subsets) of $\mathcal{B}$ is a matroid (having $\mathcal{B}$ as its set of bases). However, the downward closure of $\mathcal{B}$ is equal to $\tilde{\mathcal{F}}$, as any subset of $V$ that can be extended to a feasible solution can also be extended to a maximal feasible solution. Therefore we are left with proving (B1-B2). As we had assumed that $\mathcal{F} \neq \emptyset$, we also have $\mathcal{B} \neq \emptyset$, which establishes (B1).

For (B2), let $B_1, B_2 \in \mathcal{B}$ and $x \in B_1 \setminus B_2$. Let $c$ be the color of $x$. A simple case is when $B_2 \setminus B_1$ contains some element $y \in V_c$. Then $B_1 - x + y$ has the same number of elements of each color as $B_1$, thus it is also in $\mathcal{B}$. Now consider the other case, i.e., that $V_c \cap B_2 \subseteq B_1 - x$. Then we have

$$u_c - 1 \geq |V_c \cap (B_1 - x)| \geq |V_c \cap B_2| \geq \ell_c. \tag{1}$$

There must be another color $d$ where $B_2$ has more elements than $B_1$, for otherwise $B_2 + x$ would be feasible, contradicting the maximality of $B_2$. We claim that picking any element $y \in V_d \cap (B_2 \setminus B_1)$ yields a maximal feasible solution $B_1 - x + y \in \mathcal{B}$. The lower bounds are clearly satisfied already for $B_1 - x$ (for color $c$, this follows by (1)). The upper bound for color $c$ is satisfied by (1), and for color $d$ since $|V_d \cap (B_1 - x + y)| = |V_d \cap B_1| + 1 \leq |V_d \cap B_2| \leq u_d$. The global upper bound is satisfied as $|B_1 - x + y| = |B_1| \leq k$. To show maximality of $B_1 - x + y$, we note that any maximal set in $\mathcal{F}$ has the same size, namely $\min(k, \sum_c \min(u_c, |V_c|))$, and that $B_1 - x + y$ is already of the same size as $B_1$, which is maximal. □

## B.2 Proof of Theorem 4.2

*Proof.* The feasibility of $S$ follows as $S_{\mathcal{A}}$ is extendable and by Observation 3.2. If $\mathcal{A}$ is an $\alpha$-approximation algorithm, then it returns a solution $S_{\mathcal{A}}$ of value at least $\alpha$ times that of the best extendable set, and every feasible set is extendable. Adding elements does not decrease the value, as $f$ is monotone.

Our extra memory usage is $|\bigcup_c B_c| = \sum_c \ell_c \leq k$. □

## C Algorithms for Matroid-Constrained Submodular Maximization

In this section we describe the streaming algorithms for submodular maximization under a matroid constraint of Chakrabarti and Kale [14] (monotone $1/4$-approximation) and Feldman, Karbasi and Kazemi [28] (non-monotone $1/5.82$-approximation). We also describe how to implement FAIR-STREAMING, together with the former algorithm, so as to obtain nearly-linear runtime and oracle complexity.

Both algorithms are given access to a matroid $\mathcal{M} \subseteq 2^V$ in the form of an independence oracle. To differentiate between querying $f$ and $\mathcal{M}$, we refer to the former as oracle calls and to the latter as matroid queries.

## C.1 The monotone case

---
**Algorithm 6** Chakrabarti-Kale [14] (monotone)
---
1: $S \leftarrow \emptyset$
2: **for** every arriving element $e$ **do**
3:      $w(e) \leftarrow f(e \mid S)$
4:      **if** $S + e \in \mathcal{M}$ **then**
5:          $S \leftarrow S + e$
6:      **else**
7:          $U \leftarrow \{e' \in S : S + e - e' \in \mathcal{M}\}$
8:          $e' \leftarrow \mathrm{argmin}_{e' \in U} w(e')$
9:          **if** $w(e) \geq 2w(e')$ **then**
10:            $S \leftarrow S + e - e'$
11: **return** $S$
---

Let us look at the per-element oracle complexity and runtime. Algorithm 6 clearly makes only two oracle calls (to compute $f(e \mid S)$). As for the runtime, it is dominated by Lines 4, 7 and 8. Clearly, these can be implemented naively using $O(k)$ time and matroid queries, where $k$ is the rank of matroid $\mathcal{M}$ (we have $|S| \leq k$). The runtimes of these queries would further depend on the matroid in question.

However, for special matroids $\mathcal{M}$ the implementation can be optimized. Let us first consider the special case of $\mathcal{M}$ being the $k$-uniform matroid ($S \in \mathcal{M} \Leftrightarrow |S| \leq k$): in other words, the setting of cardinality-constrained submodular maximization. In that case, Line 4 takes $O(1)$ time, and Line 7 becomes just $U \leftarrow S$. The runtime then becomes dominated by finding the element $e' \in S$ with the lowest $w$-weight. If we maintain a priority queue $P$ containing $S$ sorted by $w$, then this can be done in $O(\log k)$ time.

Now we can extend this idea to $\mathcal{M}$ being the extendability matroid (see Definition 3.1 and Lemma 4.1) used by FAIR-STREAMING. That is, we prove Theorem 4.4. Let us restate it again for convenience.

**Theorem 4.4 (Streaming monotone)** *There exists a streaming algorithm for fair monotone submodular maximization that attains $1/4$-approximation, using $O(k)$ memory. This algorithm uses $O(\log k)$ time and $2$ oracle calls per element.*

*Proof.* Recall that FAIR-STREAMING (Algorithm 2) uses Algorithm 6 as $\mathcal{A}$. By Theorem 4.2, FAIR-STREAMING returns a feasible solution that is $1/4$-approximate. It makes $2$ oracle calls per element (these are made by Algorithm 6, see above). We are left with the runtime.

We maintain the extendability data structure from Appendix A.2. This allows us to implement Line 4 in constant time. Now let us consider the problem of finding the minimal $w(e')$ among $e' \in U$, i.e., among those elements $e' \in S$ that have $S + e - e' \in \mathcal{M}$. Clearly, whether an element $e' \in S$ is in $U$ or not depends only on its color $c'$. We will say that color $c'$ is *good* if elements $e' \in S$ of color $c'$ are in $U$. Let $c$ be the color of $e$. Following Algorithm 5, we have the following logic:

- if $t_c = u_c$, then only $c$ is good,
- otherwise, if $Q < k$ or $t_c < \ell_c$, then every color is good,
- otherwise, the good colors are $c$ and those colors $c'$ that have $t_{c'} > \ell_{c'}$.

To be able to quickly find the minimum-weight good-colored element in $S$, we will maintain a number of priority queues:

- (as before) $P$ containing $S$ sorted by $w$,
- $P_c$ for each color $c$, where we keep elements in $S \cap V_c$ sorted by $w$,
- $P'$, containing colors rather than elements: in $P'$ we keep those colors $c'$ for which $t_{c'} > \ell_{c'}$, sorted by $\min_{e' \in S \cap V_{c'}} w(e')$.

It is not hard to see that this data structure can be maintained in $O(\log k)$ time per element, and that using it we can implement the logic above in the same time. $\square$

**Our implementation** In the experimental evaluations, we use a variant of FAIR-STREAMING where the condition in Line 9 of Algorithm 6 is replaced by the more direct $f(S + e - e') \geq f(S)$. We find that this yields better solutions in practice. We still make only two oracle calls per element; this is made possible by storing the value $f(S)$ between calls. For simplicity, we also do not use the priority-queue-based data structure from the above proof of Theorem 4.4. This has no bearing on the reported experimental results, as we measure oracle calls rather than runtime.

## C.2 The non-monotone case

The non-monotone algorithm of Feldman, Karbasi and Kazemi [28], which is used by FAIR-SAMPLE-STREAMING, is similar to Algorithm 6. The main differences are that the algorithm subsamples incoming elements, and that instead of caching the marginal contribution of every element at the time it is added (as $w(e)$), it always uses the contribution of an element $e$ to the part of the current solution that arrived before $e$. For completeness, we give it as Algorithm 7.

---

**Algorithm 7** Feldman, Karbasi and Kazemi [28] (non-monotone)

---
1: $S \leftarrow \emptyset$
2: **for** every arriving element $e$ **do**
3:     with probability $2/3$ **return**
4:     **if** $S + e \in \mathcal{M}$ **then**
5:         $S \leftarrow S + e$
6:     **else**
7:         $U \leftarrow \{e' \in S : S + e - e' \in \mathcal{M}\}$
8:         $e' \leftarrow \operatorname{argmin}_{e' \in U} f(e' : S)$
9:         **if** $f(e \mid S) \geq 2f(e' : S)$ **then**
10:             $S \leftarrow S + e - e'$
11: **return** $S$

---

Here we use the notation $f(e' : S)$ to denote $f(e' \mid S')$, where $S'$ consists of those elements of $S$ that had arrived on the stream before $e'$. Note that this is different from $w(e')$ from Algorithm 6.

Algorithm 7 uses $O(k)$ oracle calls and $O(k)$ matroid queries per element.

**Our implementation**   As previously, in the experimental evaluations, in FAIR-SAMPLE-STREAMING we use a variant of Algorithm 7 where the condition in Line 9 is replaced by the more direct $f(S + e - e') \geq f(S)$. We also use $f(e' \mid S)$ in lieu of $f(e' : S)$. Finally, whenever we apply Algorithm 7 in a monotone setting, we omit Line 3.

## D   Non-monotone Streaming

### D.1   Non-monotone algorithm

We make use of the following known lemma to bound the loss in value resulting from the addition of backup elements.

**Lemma D.1 ( [7, Lemma 2.2] )**   *Let $g : 2^V \to \mathbb{R}_{\geq 0}$ be a non-negative submodular function, and let $B$ be a random subset of $V$ containing every element of $V$ with probability at most $p$ (not necessarily independently). Then $\mathbb{E}[g(B)] \geq (1 - p)g(\emptyset)$.*

**Theorem 5.2**   *Suppose $\mathcal{A}$ is a streaming $\alpha$-approximate algorithm for non-monotone submodular maximization under a matroid constraint. Then, there exists a streaming algorithm for fair non-monotone submodular maximization with expected $q\alpha$ approximation ratio, and the same memory usage, oracle calls, and running time as $\mathcal{A}$.*

*Proof.*   By assumption, we have $\mathbb{E}[f(S_{\mathcal{A}})] \geq \alpha \max_{S \in \tilde{\mathcal{F}}} f(S)$, and since $\mathcal{F} \subseteq \tilde{\mathcal{F}}$, we have $\mathbb{E}[f(S_{\mathcal{A}})] \geq \alpha f(\text{OPT})$. We define $g : 2^V \to \mathbb{R}_{\geq 0}$ to be the function $g(S) = f(S \cup S_{\mathcal{A}})$, and $B = S \setminus S_{\mathcal{A}}$ the set of backup elements added to $S_{\mathcal{A}}$. Since $B$ contains every element in $V$ with probability at most $1 - q = \max_c \frac{\ell_c}{n_c}$, then by Lemma D.1 $\mathbb{E}[g(B)] \geq q \cdot g(\emptyset)$. It follows then that
$$\mathbb{E}[f(S)] \geq q \, \mathbb{E}[f(S_{\mathcal{A}})] \geq q\alpha f(\text{OPT}).$$

□

### D.2   Non-monotone hardness

In this section we will show that our assumption that the dependence of our approximation ratio on $q = 1 - \max_{c \in [C]} \ell_c/n_c$ is necessary. Indeed, to get an approximation ratio better than $q$ for fair non-monotone submodular maximization requires nearly linear space. We prove this by reduction to the INDEX problem which we define below.

**Definition D.2**   *The INDEX problem is a two party communication problem. In it we have two parties, Alice and Bob. Alice receives $x$, a bit string of length $n$, and Bob receives a single index $i^*$ between 1 and $n$. The aim of problem is for bob to output $x_{i^*}$.*

**Theorem D.3** *[39] The one way communication complexity of index, $R_{2/3}^{pub} \geq n/100$. That is any one way communication protocol that solves INDEX on any input with probability at least $2/3$ requires at least $n/100$ bits of communication.*

We use this to prove hardness of the approximate maximization of non-monotone submodular functions under fairness constraints. Specifically we will show a reduction from INDEX to this problem.

**Theorem 5.1 (Hardness non-monotone)** *For any constant $\epsilon > 0$ and $q \in [0, 1]$, any algorithm for fair non-monotone submodular maximization that outputs a $(q + \epsilon)$-approximation for inputs with excess ratio above $q$, with probability at least $2/3$, requires $\Omega(n)$ memory.*

*Proof.* Suppose such an algorithm exists. We will produce an instance of such submodular maximization that allows us to solve INDEX with the same space complexity and success probability.

The submodular function we define will be a cut function. That is, we define some directed graph $D = (V, A)$ on the universe $V$. The function evaluated at $S \subseteq V$ will be the size of the $(S, \overline{S})$ cut. That is

$$f(S) = |\{(v, w) \in A : v \in S \ \wedge \ w \notin S\}|.$$

It is easy to see that this is indeed a non-negative submodular function.

It remains to define $V$ and $D$. Suppose Alice and Bob receive an input for INDEX for length $n$. Let the input of Alice be $x$ and the input of Bob be $i^*$. We define $V$ and $A$ based on this input

Let $a/b$ be a rational approximation of $q$ in the sense that $a, b \in \mathbb{N}$ and $q \leq a/b < q + \epsilon$. Such $a$ and $b$ can always be chosen such that $b = O(1/\epsilon)$. Let $V$ consist of three colors $V_1$, $V_2$, and $V_3$ where $V_1 = \{v_i : i \in [n], x_i = 1\} \cup \{w_i : i \in [n], x_i = 0\}$, $V_2 = \{y_{i^*}^j : j \in [b]\}$ and $V_3 = \{z^j : j \in [b]\}$. Let the color-wise constraints be $\ell_1 = u_1 = 1$, $\ell_2 = u_2 = b - a$, and $\ell_3 = u_3 = 0$, which satisfies $1 - \max_{c \in [3]} \ell_c/n_c = a/b \geq q$. If the element $u_{i^*}$ appears (that is if $x_{i^*} = 1$), it is connected to $V_3$, that is $A$ contains all edges in $\{v_{i^*}\} \times V_3$. All other elements of $V_1$ are connected to all elements of $V_2$, that is $A$ contains all edges in $V_1 \backslash \{v_{i^*}\} \times V_2$.

Alice first runs the algorithm for submodular maximization on a stream consisting of $V_1$. Since $f|_{V_1}$ is simply cardinality times $b$, Alice can answer all oracle queries without knowing Bob's input. Alice then passes the state of the algorithm to Bob, who inputs the rest of the stream: $V_2$ and $V_3$. As we show below, if $x_{i^*} = 1$, the optimal solution is $b$, while if $x_{i^*} = 0$, the optimal solution is only $a$. Therefore, Bob can correctly solve INDEX by reading off the output of the $(q + \epsilon)$-approximation algorithm, since $q + \epsilon > a/b$.

Indeed, if $x_{i^*} = 0$ and $v_{i^*} \notin V$, then

$$f(S) = |S \cap V_1| \cdot (b - |S \cap V_2|).$$

Given the strict color-wise constraints this is always equal to $b - a$. On the other hand, if $x_{i^*} = 1$ and $v_{i^*} \in V$ then we have the optimal solution

$$S = \{v_{i^*}\} \cup \{y_{i^*}^j : j \in [a]\}$$

which has value $b$.

Since INDEX needs $\Omega(n)$ memory to solve, the algorithm for fair submodular maximization must have $\Omega(n)$ memory as well. $\square$