[Reviews · NeurIPS 2020]

Review 1

Summary and Contributions: This submission formulates algorithms for submodular optimization in a streaming setting subject to fairness constraints. This is the first work studying algorithmic fairness in this context. Algorithms are provided for both cases of monotone and non-monotone objective. An empirical evaluation is provided.

Strengths: + Novel and interesting problem setting + In monotone case, no loss in approximation factor over "unfair" setting + In non-monotone case, there can be a significant loss in approximation, but authors provide a hardness result that shows their approach is within constant factor of optimal.

Weaknesses: - Results in Section 3 and 4 appear to follow trivially from Lemma 4.1. Once we know that the "fair-extendable" constraint forms a matroid (which is not difficult to show), the results follow directly. - In the non-monotone case, the results do not follow directly from Lemma 4.1, as extending a solution may produce an arbitrarily bad one. But the algorithm still differs little from the unfair case, with the only addition being the reservoir sampling.

Correctness: Appears to be. Did not check completely.

Clarity: The paper is well written.

Relation to Prior Work: Related work discussion and empirical comparison is sufficient.

Reproducibility: Yes

Additional Feedback: The theoretical depth of the contribution seems somewhat lacking. As discussed above, the main results rely heavily on previous work (streaming algorithms for unfair matroid constraint). Even in the non-monotone case, extending to a fair solution does not seem very difficult (while it is trivial in the monotone case). Perhaps fairness constraints that did not lie within a matroid would be more interesting. I would also be interested in a bound between the fair and unfair instances. I.e. the cost of fairness. The empirical justification that this is small may be highly application / parameter dependent. It is possible that I am incorrect and have missed something, in which case the authors should correct me during the feedback period. ===== edit after rebuttal ===== After reading the rebuttal, I find the following argument convincing: As Celis et al. [12] didn't realize the constraints lie within a matroid, the contribution of this paper includes making this observation, which was not obvious. Further, I think the hardness result of the authors makes the concept of excess ratio a bit more interesting than I had previously assessed, and it could perhaps be applied in other settings. I have raised my score from '5' to '6'.


Review 2

Summary and Contributions: This paper studies streaming submodular maximization under fairness constraints. Importantly these are not matroid constraints because they contain lower bounds on the number of elements from each protected group. The authors propose the concepts of extendability and excess ratio, as well as several black box reductions to previous algorithms which eliminate fairness violations while optimally preserving performance guarantees

Strengths: Relevance/Significance: Overall I think this a good paper that posits and resolves a natural open problem given previous work. Notions of extendability and excess ratio are potentially of independent interest. Experiments are thorough and show a clear tradeoff between objective value, fairness violations, and running time

Weaknesses: Novelty: My biggest concern is that this paper borrows heavily from previous work on streaming submodular maximization (CK and FKK theory/algorithms). While useful, the key contributions (the family of extendible sets is a matroid, maintain a small set of backup elements) are somewhat incremental given the previous algorithms

Correctness: Proofs appear to be correct, and experiments are sound

Clarity: - In general the text flows well and is a pleasure to read - I appreciate running many baselines, but it is often difficult to distinguish between them in the figures - Please clarify that in Figure 1[c,f,i] the y axis is the number of oracle calls at iteration k

Relation to Prior Work: All relevant prior work is discussed, attributing parts of the proposed method to previous results where appropriate

Reproducibility: Yes

Additional Feedback: Questions: - How much do the sets returned by these algorithms differ? Do certain solutions have large intersection with each other? - Do the results assume the number of colors C is constant? It would be better if the running times and memory requirements depended on this quantity. --------- EDIT: The authors addressed all my questions in their feedback and made a strong case for the paper's novel contributions. So I am keeping my score at accept


Review 3

Summary and Contributions: The authors study submodular maximization subject to a cardinality constraint and additional fairness constraints in the streaming setting. Fairness here is interpreted as follows: the universe is partitioned into disjoint groups and we are given lower and upper bounds on the elements that should be chosen from each group. As the authors show, the problem can be reduced to submodular maximization subject to a matroid constraint. In the monotone case there is no loss in the approximation, while in the non-monotone case the approximation factor is parametrized by the so called excess ratio. Finally, the authors experimentally evaluate their algorithms on various applications related to ML.

Strengths: This is an interesting twist on a classic problem in discrete optimization that has lately found applications in ML and other domains with massive data. The main idea, although simple, is novel and the results seem sound, at least to the extent that I checked. The paper is definitely relevant to the NeurIPS community.

Weaknesses: Although I think this is a very nice work, I am afraid that the technical contribution is low for NeurIPS. The main ideas are cute but rather straightforward as are their proofs. On a high level, Sections 3 and 4 boil down to a special case of a matroid constraint plus some irrelevant elements, and then 5.2 is a direct application of a lemma of Buchbinder et al.

Correctness: The results seem correct, at least to the extent that I checked.

Clarity: The paper is very well-written and it was a pleasure to read.

Relation to Prior Work: The paper clearly discussed how it differs from previous contributions.

Reproducibility: Yes

Additional Feedback: -- How is it possible that Upper-bounds makes more calls? Isn’t it just the FKK algorithm without the extendability requirement? ** Update after reading the other reviews and the authors' responses ** While I still think that the technical contribution is the main weakness of this work, I agree with the authors that simplicity should not be penalized. So I raise my score to a weak accept.


Review 4

Summary and Contributions: The paper studies the problem of streaming submodular maximization subject to a cardinality constraint and a so-called “fairness” constraint, which is like a laminar matroid, but also includes lower bounds on number of elements from a partition. From a submodular optimization point of view, this “fairness” constraint is challenging because it is not down-closed. The main contributions are several adaptions of existing submodular streaming algorithms to accommodate the fairness constraints. In the case of non-monotone objectives, the approximation depends on an “excess ratio”. Authors demonstrate that this excess ratio is inherent to the problem by proving hardness results which also depend on the excess ratio. Finally, experiments are run on benchmark datasets.

Strengths: The strength of the work is the development of appropriate reductions, which allows for existing submodular streaming algorithms to accommodate the fairness constraints, with minor modifications. These reductions are simple and clean and the fact that they can be easily incorporated into existing frameworks is, from an algorithmic point of view, very nice. Another strength of the work is the clarity of the writing itself. The paper is quite pleasant to read and the results are easy to understand and verify the high level ideas.

Weaknesses: One limitation -- which is a limitation of nearly any technical response to algorithmic bias and unfairness -- is that there are scenarios in which the algorithm itself may actually inflict more unintended unfairness. The limitation I want to raise here is that, in this work, such unintended harm may be caused precisely by the reduction techniques which make the algorithmic ideas elegant and simple. Before continuing, I want to remark that this does not affect my score at all, but I do want to bring it to the authors attention, as they may find this sort of example interesting. Consider applying Algorithm 2 (for the monotone setting) to college admissions, where we aim for our admissions to have sufficient representation across racial and cultural groups. Algorithm 2 admits students the same way it was going to admit without the fairness constraints, while saving room for minorities **at the end**. Then, at the end, the admissions are “augmented” with a few reserved minority students. In this sense, the algorithm doesn’t actively try to incorporate all racial / cultural groups from the beginning. Instead, the algorithm makes offers as usual and then stuffs in a few minority offers in at the end, to meet a quota. If these actions are observed by the applicants, it could cause further harm. The real problematic part of this example is the fact that we used a reduction, which was algorithmically elegant but perhaps not the right tool to use given this context.

Correctness: I did not check any of the proofs in the appendix, but the details presented in the main body give me confidence that all results are correct. The experimental methodology appears sound.

Clarity: The paper is overall very well written. Beyond the abstract and introduction, the paper is very clear and is a wonderful read. The presentation guides the reader through the results in an intuitive way. The details in the analysis are reserved for the appendix, but this is okay because the main ideas are present in the paper. Upon my first reading, I did feel rather negatively about how the abstract and introduction frame the problem, especially with respect to “fairness”. In particular, I draw attention to the section in the abstract “Is it possible to create fair summaries for massive datasets? We give an affirmative answer by developing the first streaming algorithms for submodular maximization under fairness constraints”. Surely, the authors must recognize that this work alone - a particular technical methodological contribution in the grand scheme of things - does **not** address the larger non-technical and societal question of fair summaries in massive datasets. The introduction writes about “fairness constraints” with an authoritative tone, as if this mathematical formulation somehow captures the essence of fairness. This tone in writing at the beginning of the paper did turn me off quite a bit until I read further. To their credit, the authors do acknowledge that “the fairness notion [they] consider...does not capture other notions of fairness” and “no universal metric of fairness exists”. I think that the paper would benefit from using this framing from the beginning, rather than saving it until the end.

Relation to Prior Work: The prior work of Celis et al on this fairness constraint is well described. However, there are some minor errors in the prior work on streaming submodular maximization. First, the paper [15] was the first to develop non-monotone maximization over a matorid, not [46]. Moreover, the paper [46] has been shown to be incorrect (you can see details in Haba et al "Streaming Submodular Maximization Under a $k$-system constriaint" which appeared in ICML 2020). So, it is appropriate to simply cite [28]. Finally, I would add that there has been prior work on fairness in submodular maximization, although the formulation is different. That is "Scalable Deletion-Robust Submodular Maximization: Data Summarization with Privacy and Fairness Constraints" by Kazemi et al ICML 2018. I should also point out that the work of [28] (Feldman et al “Do less, get more: streaming submodular maximization with subsampling” NeurIPS 2018) has since been improved to get tight memory complexity and also work in parallel settings. It might be worth citing this improved result, as it further improves the algorithms presented in this paper and perhaps even extends them to the parallel (low-adaptivity) setting too. The paper is by Kazemi et al titled “Submodular streaming in all its glory: tight approximation, minimum memory, and low adaptive complexity” and appears in ICML 2019.

Reproducibility: Yes

Additional Feedback: As I mentioned before, I would encourage authors to change the initial framing of the paper, which promises to “giv[e] affirmative answers” to the question of whether “it is possible to create fair summaries for massive datasets”. I think this part of the abstract might be better served describing the fairness constraint, so that the reader better understands this aspect. Some very minor typo points: 1. There also seems to be some odd formatting with bold and numbered text in the first paragraph on Section 6. 2. On Line 22, the sentence “submodularity is a natural way to capture the diminishing returns property of set functions, which holds for a variety of machine learning problems” is slightly awkward and misleading as written. It appears as if all set functions have an inherent diminishing returns property and that submodularity is capturing this. Of course, fixing this is just a matter of rephrasing. 3. I recommend not using the word “bias” in Line 317, as it is not used in this context earlier in the paper. ------------------------ EDIT ----------------------- I thought this was a very nice paper and I didn't have many concerns. Authors sufficiently responded to the concerns I had. I still maintain my score of accept. This is a minor unrelated point, but I feel that I need to make it: perhaps as you can tell by my review, I am very critical of algorithmic fairness literature. I think that the greatest harm this literature poses is actually legitimizing new forms of algorithmic harm by introducing mathematical constructs which are seen as "objective" and "true". In particular, I **really** want to push back against this notion of "the price of fairness". It appears in the Celis et al paper and I wish it had never been stated. At first, it might seem (especially to folks in the algorithms community) as a natural concept: how does the objective value decrease when more constraints are added? However, I want to argue that in the context of societal fairness in algorithmic decision making, this notion is incredibly backwards and actually quite harmful. Let's consider the application of job placement. If interpreted in this context, the "the price of fairness" says that you could have a really awesome workforce if you didn't have to care about diversity but then once you decide to care about diversity, you pay a cost of a less desirable workforce. This is totally bonkers because, presumably, we incorporated diversity into our hiring model exactly because we believed that such diversity was intrinsically valuable, not because it was "a constraint to satisfy". So, I really want to argue for us to stop using this term "price of fairness" since it reinforces the idea that diversity is a cost we incur rather than something that we gain. Sure, it's moderately interesting optimization question, but once it's connected to actual applications it seems to be a very limiting and backwards concept. I hope this point comes across as genuine concern and not that I'm, like, virtue signaling from a morally high soap box. Because that's what twitter is for, not CMT reviews :) I'm not necessarily arguing that "price of fairness" be removed from the paper, but perhaps I am suggesting that the utility of this term could be discussed a little bit in this paper and also that, in future work, we do not give this concept too much credibility.

[Author Response · NeurIPS 2020]

We would like to thank the reviewers for their thorough reading of the paper and their comments. Below we address some of the main points. We will address all comments in the final version.

**Technical contribution and novelty:** All reviewers seem to agree that the problem we address is important and timely, and that our results are interesting and useful. We believe that the fact that we achieve these results in a simple way should not be held against us. (Ideally, it should not be the case that a paper achieving the same results with a more complex solution would have a higher chance of being accepted.) On the contrary, we think that, as Reviewer 4 pointed out, the simplicity of the reductions that we have developed, which allow for the incorporation of fairness constraints easily into existing frameworks without significant changes, is one of the strengths of our work, and it makes our algorithms easy to apply in practice.

In terms of novelty of the work and theoretical/technical contribution, we would like to clarify several points.

• The fairness constraints do not themselves constitute a matroid and the natural reduction to submodular maximization subject to matroid constraints does not work. Our reduction relies on the novel notion of *extendable* sets and the observation that the family of extendable sets forms a matroid. Although this observation is not difficult to prove once stated, the idea itself is far from intuitively obvious.
• Theorem 4.3 follows directly from our reduction (Theorem 4.2), but this is not the case for Theorem 4.4 (the main result in the monotone case). Achieving the faster running time in Theorem 4.4 requires more work. In works on matroid-constrained submodular maximization, the complexity is given in terms of the number of matroid queries. In the case of the matroid of extendable sets, a naive implementation of such a query would create dependencies on the number of colors (the value $C$) for the running time. Therefore we designed an implementation using efficient data structures that handles such queries in $\widetilde{O}(1)$ time for each element of the stream (see Appendices B.2 and D).
• The *excess ratio* concept we introduce in this work is novel. Without it, one cannot apply the previous ideas to the non-monotone case. This concept may also be of independent interest for various classic problems in the streaming setting subject to fairness constraints, e.g., matching, vertex cover, prophet, or secretary problems.
• One of the main contributions of this work is the hardness result (Theorem 5.1), showing that our non-monotone algorithm is asymptotically tight. This also confirms the theoretical importance of the *excess ratio*. This result is novel, and its proof is fairly involved (see Appendix E.2).

**Reviewer 1:** We hope that the points above address your concerns regarding the theoretical contribution of our work.
• *"I would be interested in a bound between the fair and unfair instances."* From a theoretical point of view, one can come up with a simple example where the optimum "fair" solution has value zero, while the optimum "unfair" solution has value one. Hence, the cost of fairness is in general unbounded, but as you correctly mentioned it is indeed highly application dependent. To account for this, we ran experiments for 4 applications, with different datasets and parameters.

**Reviewer 2:** We hope that the points above address your concerns regarding the novelty of our work.
• *"Please clarify that in Figure 1[c,f,i] the y axis is the number of oracle calls at iteration k"* The $y$-axis corresponds to the total number of oracle calls for a given cardinality constraint $k$. We will emphasize this in the final version.
• *"How much do the sets returned by these algorithms differ?"* In some cases they differ a lot. As reported in Figures 1-3, the solutions returned by the "unfair" algorithms violate the fairness constraints significantly, and accordingly they must also differ significantly from the solutions returned by our proposed algorithms.
• *"Do the results assume the number of colors $C$ is constant?"* No, we do not make such an assumption; the number of colors can be any value between $1$ and $n$ (length of the stream). Surprisingly, the running time and memory consumption are independent from the value of $C$. We achieve this by utilizing efficient data structures and update techniques. The details are provided in the appendix.

**Reviewer 3:** We hope that the points above address your concerns regarding the technical contribution of our work.
• *"How is it possible that Upper-bounds makes more calls? Isn't it just the FKK algorithm without the extendability requirement?"* This is a great point. Notice that in the case when we have only upper-bounds, there are more feasible elements that we need to check for swapping compared to the case where we have both upper and lower bounds. Therefore, Upper-bounds can require more oracle calls.

**Reviewer 4:** Thank you for the insightful comments and constructive feedback. We will adjust the tone of both the introduction and abstract, and correct the errors in the prior work section. Regarding the limitation illustrated with the college admission example: Note that Algorithm 2 will admit students, from the beginning, differently compared to an algorithm without fairness constraints, due to the extendability constraints. Moreover, we find that in practice, augmentation is not required, as the cardinality constraint is reached by the time the stream is finished.

[Meta-Review · NeurIPS 2020]

All the reviewers liked the idea of the paper. Also, the authors had a successful rebuttal that convinced us this is a paper worthy of acceptance. I also agree with the authors that the simplicity of their proposed algorithm should be embraced.